# Secure AI With Data Wallets: Privacy-Preserving Solid Architecture for Personal Data LLMs

### Abstract

This paper addresses the challenge of enabling highly personalized AI interactions with personal data storage, while preserving confidentiality, integrity and availability. We propose a novel architecture that leverages Trusted Execution Environments (TEEs) of confidential computing to create a secure processing layer between decentralized Solid personal data stores (Pods) and AI services. Our approach ensures that AI models can process sensitive user data while guaranteeing that neither service providers nor infrastructure operators can access the raw data or inference content. We demonstrate how this architecture can be implemented using remote attestation, end-to-end encryption, and hardware-based isolation, creating a verifiable trust chain from the user's personal data storage to the AI processing environment. The resulting system enables personalized AI services without compromising on data sovereignty principles that are core to the personal safety inherent in a Solid ecosystem.

**Keywords:** Solid, Personal Data Store, Data Wallet, Trusted Execution Environment, Confidential Computing, Privacy-Preserving AI, Data Sovereignty, Agentic

## 1   Introduction

Personal data wallets represent a paradigm shift in digital information management, enabling users to store their data in user-controlled repositories rather than only within service provider backend systems. The Solid protocol has pioneered this approach through its Pod standard, providing decentralized storage with granular access control.

Concurrently, AI systems increasingly offer personalized experiences that typically require access to sensitive user data for processing. This creates a fundamental tension: effective personalization demands more personal data, yet individuals are increasingly wise to fight against surrendering control of their information.

As data wallets rapidly gain prominence, a critical question emerges: can AI systems provide personalized services while using the wallet concepts to preserve data privacy and user control? Specifically, how can AI interact with Solid Pods as data wallets without exposing sensitive information to third parties?

This paper presents a working architecture that addresses this challenge by combining Solid's data governance with confidential computing techniques. We create a secure processing layer between Solid Pods and AI services, ensuring data remains encrypted throughout the inference process. Through Trusted Execution Environments (TEEs), remote attestation, and end-to-end encryption, we establish a verifiable trust chain enabling personalized AI services without compromising data sovereignty.

Key contributions include:

- A comprehensive architecture for privacy-preserving AI interactions with Solid Pods

- A secure data flow design preventing raw data access by service providers and infrastructure operators

- An implementation approach leveraging confidential computing and remote attestation

- Analysis of security properties and privacy guarantees of the proposed system

# 2 Background and Related Work

## 2.1 Solid for Personal Data Storage

Solid provides a user-centric architecture that gives individuals control over their digital information through "Pods" (Personal Online Data Stores) with granular access control mechanisms. Its key innovation is separating data storage from applications, enabling users to grant and revoke access without migrating their data—a significant shift from traditional models where service providers maintain custody of user information.

## 2.2 Privacy Challenges in AI Interactions

AI personalization typically requires access to raw user data, creating privacy challenges including: (1) data exposure to service providers, (2) loss of user control over data usage, (3) potential data extraction through model memorization, and (4) infrastructure operator access risks. While techniques like federated learning, differential privacy, and homomorphic encryption address some concerns, they often introduce significant trade-offs in performance, accuracy, or usability.

## 2.3 Confidential Computing and TEEs

Trusted Execution Environments (TEEs) provide hardware-based isolation where data remains encrypted in memory except within the secure environment. Recent advancements include widespread cloud provider support for confidential VM instances, enhanced remote attestation capabilities, and extension to AI accelerators. Confidential computing addresses limitations of purely cryptographic approaches to privacy-preserving AI by maintaining security guarantees with practical performance.

# 3 System Architecture

Our proposed architecture establishes a secure bridge between user-controlled Solid Pods and AI services through a multi-layered approach to privacy and security. Figure 1 illustrates the overall architecture, highlighting the three primary layers: the User Domain, the Secure Processing Layer, and the AI Service Layer.

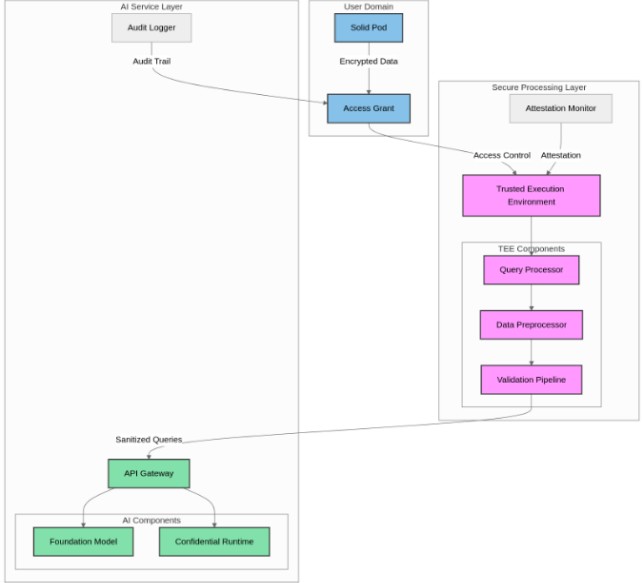

Figure 1: Three-layer privacy-preserving architecture showing the secure data flow between user-controlled Solid Pods and AI services. Pink components represent secure processing elements, blue components indicate data storage and access control, and green components show AI service infrastructure. The architecture ensures data remains encrypted outside of a verified Trusted Execution Environment.

## 3.1 User Domain

At the foundation of our architecture is the User Domain, which consists of:

- **Pods:** Secure storage with built-in access control mechanisms
- **Access Control:** Manages permissions and enforces granular access policies
- **Client Proxy:** A user-side component that handles attestation verification and encryption

The User Domain maintains complete control over data access. No data leaves this domain without explicit authorization and end-to-end encryption.

## 3.2 Secure Processing Layer

The Secure Processing Layer provides a protected bridge between user data and AI services through a Trusted Execution Environment (TEE). This layer includes:

- **Query Processor:** Handles incoming AI requests and manages access to user data
- **Data Preprocessor:** Transforms raw data into AI-ready formats while preserving privacy
- **Validation Pipeline:** Ensures that processed results maintain privacy before returning to the user
- **Attestation Monitor:** Continuously verifies the integrity of the TEE environment

The TEE provides hardware-level isolation for all data processing. Memory encryption protects data during processing, while remote attestation verifies the environment's integrity.

## 3.3 AI Service Layer

The AI Service Layer provides the actual AI capabilities while remaining isolated from raw user data:

- **API Gateway:** Manages requests to the AI service
- **Foundation Model:** Provides core AI capabilities without access to unencrypted user data
- **Confidential Runtime:** Ensures model execution occurs within a protected environment
- **Audit Logger:** Maintains comprehensive records of all AI operations without storing sensitive data

## 3.4 Secure Data Flow and Processing

The data flow within our architecture follows a carefully designed path that preserves privacy at each step:

1. User authorizes AI access to specific Pod resources
2. Client proxy verifies the TEE through remote attestation
3. Upon successful verification, the client establishes encrypted communication with the TEE
4. The client sends encrypted data and queries to the TEE
5. Inside the TEE, data is decrypted and preprocessed for AI consumption
6. AI inference occurs within the confidential runtime
7. Results are validated to ensure they contain no sensitive information leakage

8. Processed results are encrypted and returned to the user

This workflow ensures that:

- Data remains encrypted except within the verified TEE
- Service providers cannot access unencrypted user data
- Infrastructure operators have no access to data or processing details
- All operations are logged for audit purposes without compromising privacy

# 4 Implementation Approach

Our implementation approach builds on techniques demonstrated in production confidential computing systems. While these focus on securing general AI inference services, our architecture adapts similar attestation and encryption patterns specifically for the Solid ecosystem, where data ownership and granular access control are fundamental principles. Implementing our architecture requires integrating Solid Pods into TEEs through a three-phase approach:

## 4.1 Infrastructure and Security Configuration

The foundation begins with enhanced Solid Pod configurations featuring granular access controls that limit data exposure to only necessary resources. We utilize confidential computing environments that provide hardware-level memory encryption and integrity protection. Communication between components occurs through TLS 1.3 channels with perfect forward secrecy.

The security layer implements hardware-based memory encryption within the TEE and establishes secure key management using hardware-based storage that prevents extraction of cryptographic materials. Remote attestation functions as the critical security component, generating cryptographic proof of the environment's integrity before any data sharing occurs. This creates a verifiable chain of trust from hardware root to application layer, with measurements validated against known-good values by a Coordinator component.

## 4.2 AI Integration and Privacy Safeguards

AI integration maintains privacy through several mechanisms. First, query sanitization prepares data for AI consumption while removing unnecessary sensitive information, all within the TEE's protected boundaries. Foundation models run in confidential runtimes loaded from verified sources with integrity checks. For GPU acceleration, we leverage confidential computing capabilities of modern AI accelerators, maintaining the trust chain across hardware boundaries.

Response validation serves as the final privacy safeguard, implementing automated checks for data leakage and statistical techniques to prevent model inversion attacks. Throughout implementation, we balance security with performance through optimized processing, strategic caching, and efficient communication protocols—delivering practical performance without compromising privacy guarantees.

# 5 Privacy Guarantees and Verification

Our architecture provides verifiable privacy guarantees through both technical and procedural means, establishing trust and ensuring privacy protections work as intended.

## 5.1 Security Model

The security model rests on four foundational assumptions: (1) hardware TEE implementations provide genuine isolation properties; (2) remote attestation correctly verifies TEE integrity; (3) encryption algorithms follow cryptographic best practices; and (4) users maintain proper access control over their Solid Pods.

Under these assumptions, our system delivers four essential privacy properties: confidentiality (raw data never accessible outside the TEE); integrity (data cannot be modified by unauthorized parties); verifiability (users can confirm security properties before sharing data); and sovereignty (users maintain control throughout the data lifecycle).

## 5.2 Remote Attestation Process

Remote attestation provides cryptographic proof that the system is in a known-good state before data processing begins. The process starts when the client requests attestation from the TEE, initiating a challenge-response protocol that prevents replay attacks. The TEE generates evidence of its current state—including hardware configuration, software measurements, and runtime environment details—which is cryptographically signed using hardware-based keys anchored in the processor's secure elements.

The client verifies this signed evidence against known-good values before proceeding with data exchange. The attestation process repeats periodically during long-running sessions to guard against time-of-check-to-time-of-use attacks, maintaining the verification chain throughout the data processing lifecycle.

## 5.3 Audit and Verification

Our architecture implements audit and verification mechanisms that provide ongoing assurance without compromising security. Comprehensive logging captures significant operations without recording sensitive data contents, using privacy-preserving techniques such as data minimization and pseudonymization. These logs are cryptographically protected to prevent tampering.

Verification tools allow independent assessment of security properties for both users and authorized third parties, following the principle of "trust but verify." Transparency reports summarize key metrics without exposing sensitive details, documenting attestation success rates, system updates, and security incidents. Together, these mechanisms create accountability while preserving the core privacy guarantees.

# 6 Discussion

## 6.1 Benefits of the Approach

Our architecture offers key advantages over traditional AI personalization approaches while maintaining practical utility. The primary benefit is enhanced privacy through data sovereignty—users receive personalized AI services without surrendering control of their data, as processing occurs only within verified secure environments. This fundamentally transforms the trust model from one based on provider promises to one with verifiable technical guarantees.

The architecture provides cryptographic verification through remote attestation, allowing users to verify the security state of processing environments before sharing data. This creates auditable proof of the system's trustworthiness beyond the traditional "black box" approach. Additionally, the system aligns with regulatory frameworks like GDPR and the Data Governance Act through its data minimization, user control mechanisms, and transparent processing.

Security is enhanced through a reduced attack surface, with data remaining encrypted throughout most of its lifecycle and decryption limited to verified secure environments. This defense-in-depth approach substantially reduces breach impact, protecting data even if cloud infrastructure is compromised.

## 6.2 Limitations and Future Work

Despite its advantages, several limitations remain. Performance overhead (typically 10-30%) of isolation and encryption mechanisms impacts the cost calculation for deploying real-time applications. Hardware dependencies constrain deployment flexibility, as the architecture requires specific TEE technologies not universally available across computing platforms. Current hardware has a model size limitation due a single-GPU code constraint, expected to be significantly improved in 2025 by multi-GPU code.

Implementation challenges include attestation complexity, requiring careful key management and protection against various attacks. Perhaps most challenging is creating user-friendly security verification—communicating complex security guarantees to non-specialist users in comprehensible ways remains an open problem.

Our future work will address these limitations through: (1) performance optimization via specialized hardware acceleration and selective TEE boundary use; (2) support for more complex AI workflows including federated learning approaches; (3) improved security verification usability through intuitive interfaces; and (4) integration with emerging standards for data wallets and digital identity to ensure broader ecosystem compatibility while maintaining strong privacy guarantees.

# 7    Conclusion

This paper has presented a novel architecture for enabling privacy-preserving AI interactions with personal data stored in Solid Pods. By leveraging Trusted Execution Environments, remote attestation, and end-to-end encryption, our approach allows users to benefit from personalized AI services without compromising data sovereignty.

The proposed architecture demonstrates that the apparent tension between personalization and privacy can be resolved through careful system design and modern confidential computing techniques. As personal data wallets become more prevalent in the digital ecosystem, such privacy-preserving approaches will be essential for maintaining user trust while enabling innovative AI applications.

Our work contributes to the ongoing development of the Solid ecosystem by addressing one of the key challenges in personal data management: how to safely derive value from sensitive personal information without surrendering control or compromising privacy. By providing a blueprint for secure AI integration with Solid Pods, we hope to advance the adoption of user-centric data management practices while enabling the benefits of AI personalization.

