# OpenReview forum: "Secure AI With Data Wallets: Privacy-Preserving Solid Architecture for Personal Data LLMs"
_SolidProject.org/SoSy/2025/Privacy_Session — SoSy2025-Privacy_

### Official Review · ~Anelia_Kurteva1 · 2025-03-20
**Novel application of TEEs for SOLID to facilitate secure AI. Interesting work with potential.**

**Rating:** 7
**Confidence:** 4

**Review:**

The paper presents a quite interesting application of PETs, namely TEEs for secure AI in SOLID. I think looking into PETs is a logical next step for advancing AI in decentralised settings. Following this, I think the paper is a valuable piece of work I would be happy to hear more about on the event.

Some comments that I think can significantly improve the quality of the paper:

Better justification of why exactly TEEs are used. Why was this the selected PET the authors explored first? Were there any other considerations and requirements that led the system design decisions?

I expected to see at least some referencing of the stated facts  and to previous work on SOLID.

Figure 1 is useful for the reader, however, I would add numbering to the process flow. A more formal language for diagrams (e.g. UML) could have been utilised. Later on in 3.1 the author mentions the user domain is at the core of the architecture, however this contradicts the presented diagram on Figure 1 where the user domain is in the middle. Are the layers hierarchical etc.? Access control is presented as a process but from the text it seems as it is a separate component. The figure should better reflect the layered architecture as now it is a bit misleading.

"The TEE provides hardware-level isolation for all data processing." This needs more elaboration as now it seems the TEE is mostly software-related.

Section 3.4 is useful but can be a process flow diagram instead.

How is data encrypted exactly? What is the approach? (step 4 in 3.4)

"All operations are logged for audit purposes without compromising privacy". What does this mean actually? Is this because data is encrypted, the audits can be accessed by authorised entities, sanitisation, used only based on one's consent etc.?

Missing information on what foundation models were used to test the work. Does this approach work will all current foundational models? What are the limitations here?

Unclear what the authors mean by known-good values. Giving some examples should help the reader.

I would be interested to know more about these key metrics that help summarise transparency.

I agree with the authors regarding the conclusions and challenges ahead. Looking forward to what comes out of this work!

---

### Official Review · ~Jesse_Wright1 · 2025-03-22
**Solution architecture using TEEs to fetch and run inference over data from a Pod**

**Rating:** 6
**Confidence:** 5

**Review:**

This paper describes a solution architecture in which uses TEE’s to perform AI inference over data that has been collected from a Solid Pod; ensuring that the raw data from within the Pod is not made available to the requestor - just the result of the inference.

The architecture described is a sensible solution architecture to that compliments the PET capabilities of TEE’s and Solid. However, this paper lacks code artefacts, and any insight into novel additions to this solution architecture including:
 - How query santisation is performed,
 - How response validation is performed, and
 - What foundation models have feasibly been run within this architecture

Without these supplementary materials and insights this paper provides little that can be built upon beyond the suggestion to use TEE’s and Solid in a solution architecture. We request the author add these materials and insights to improve the impact of their contribution. Further requested insights, clarifications, fact-checking and substantiation of claims are described below.

> Personal data wallets represent a paradigm shift in digital information management, enabling users to store their data in user-controlled repositories rather than only within service provider backend systems. The Solid protocol has pioneered this approach through its Pod standard, providing decentralized storage with granular access control.

This statement is somewhat misleading. Solid Pods happen to be an appropriate holder service within the data wallet ecosystem. I would avoid labelling Solid Pods as data wallets - please see https://blog.jeswr.org/2025/02/14/data-wallets which discusses this topic.

This paper also does not appear to in any way use or require Verifiable Credentials so I would recommend removing the terminology of Data Wallets from this paper.

> Trusted Execution Environments (TEEs) provide hardware-based isolation where data remains encrypted in memory except within the secure environment.

Please improve the accuracy of this description - suggest using definition closer to https://en.wikipedia.org/wiki/Trusted_execution_environment

> AI integration maintains privacy through several mechanisms. First, query sanitization prepares data for AI consumption while removing unnecessary sensitive information, all within the TEE’s protected boundaries.

Please describe this query sanitsation and if possible make the code available.

> For GPU acceleration, we leverage confidential computing capabilities of modern AI accelerators, maintaining the trust chain across hardware boundaries.

Please identify what confidential computing capabilities you are describing

> Response validation serves as the final privacy safeguard, implementing automated checks for data leakage and statistical techniques to prevent model inversion attacks.

Please describe this response validation and make this code available.

> Throughout implementation, we balance security with performance through optimized processing, strategic caching, and efficient communication protocols—delivering practical performance without compromising privacy guarantees.

Please describe or provide access to the components such as the strategic caching - without this, these statements can neither be verified, or be useful to readers of this paper.

> Additionally, the system aligns with regulatory frameworks like GDPR and the Data Governance Act through its data minimization, user control mechanisms, and transparent processing.

Has this been legally validated?

>  (1) performance optimization via specialized hardware acceleration and selective TEE boundary use;

This is not a particularly useful discussion without some kind of empirical performance analysis of the current system

> (2) support for more complex AI workflows including federated learning approaches;

Please justify the benefit that federated learning would provide. For instance, with a use case.

> (4) integration with emerging standards for data wallets and digital identity to ensure broader ecosystem compatibility while maintaining strong privacy guarantees

Please describe the kind of integration

> TLS

Are you doing any novel e2ee or just relying on HTTP(S) encryption. Note that HTTP(S) is not e2ee as e2ee is usually describing full client-client encryption.


> Questions


 - 3.4.1 What AI is being used here - and can feasibly be used with TEEs available to consumers. At present AI does not appear to be a necessary part of the solution architecture - I would suggest saying “data processing in the TEE - which may be AI inference”.
 - 3.4.4 How does the client know that data is being sent to a TEE
 - 5.2 How is remote attestation being described in the solution that you have built.

> Requested additions to further improve this contribution

 - Make code available
 - Describe a use case that this architecture has been applied to

---

### Decision · Program_Chairs · 2025-04-01

Accept